# Exploiting a New Approach to Destroy the Barrier of Tumor Microenvironment: Nano-Architecture Delivery Systems

**DOI:** 10.3390/molecules26092703

**Published:** 2021-05-05

**Authors:** Yanting Sun, Yuling Li, Shuo Shi, Chunyan Dong

**Affiliations:** 1Department of Oncology, Shanghai East Hospital, Shanghai Key Laboratory of Chemical Assessment and Sustainability, School of Chemical Science and Engineering, Tongji University, Shanghai 200120, China; sunyanting0210@163.com; 2Department of Pharmacy, Shanghai East Hospital, Tongji University School of Medicine, Shanghai 200120, China; yuling19893@163.com

**Keywords:** tumor microenvironment, targeted therapy, nanoparticles, nano therapeutics, tumor imaging

## Abstract

Recent findings suggest that tumor microenvironment (TME) plays an important regulatory role in the occurrence, proliferation, and metastasis of tumors. Different from normal tissue, the condition around tumor significantly altered, including immune infiltration, compact extracellular matrix, new vasculatures, abundant enzyme, acidic pH value, and hypoxia. Increasingly, researchers focused on targeting TME to prevent tumor development and metastasis. With the development of nanotechnology and the deep research on the tumor environment, stimulation-responsive intelligent nanostructures designed based on TME have attracted much attention in the anti-tumor drug delivery system. TME-targeted nano therapeutics can regulate the distribution of drugs in the body, specifically increase the concentration of drugs in the tumor site, so as to enhance the efficacy and reduce adverse reactions, can utilize particular conditions of TME to improve the effect of tumor therapy. This paper summarizes the major components and characteristics of TME, discusses the principles and strategies of relevant nano-architectures targeting TME for the treatment and diagnosis systematically.

## 1. Introduction

Owing to the complex and continuously evolving tumor microenvironment (TME), cancer becomes one of the most difficult diseases to cure all over the world. The bidrectional interactions between tumor and the TME bring about the progression, therapeutic resistance, and metastasis of cancer [1]. TME composes various supporting cells such as immune cells, fibroblasts, endothelial cells, and extra components like exosomes, cytokines, enzymes, growth factors, and extracellular matrix (ECM), etc. [2,3]. In addition, the tumor microenvironment displays unique pH values, hypoxic condition, high ATP concentration, and abundant tumor microvasculature [4,5,6]. The communications between tumor cells and the microenvironment result in drug resistance by changing the phenotypes of tumor cells as well [7]. Therefore, treatment targeting the microenvironment has attracted increasing attention.

The rapid development of nanotechnology has provided a good platform for early diagnosis and more effective therapy of tumors [8]. Nanoparticles (NPs) can effectively improve the pharmacokinetic and pharmacodynamics properties of drugs and improve the therapeutic effect due to its special size, shape, and material [9]. Coated with folic acid, hyaluronic acid, and other molecules, nanoparticles can be used as good carriers concentrating drugs at the tumor site much better. Due to the high biocompatibility, good targeting property and low toxicity of organic nanomaterials, related materials have been developed in large quantities. Some organic nanomaterials like liposomes (pegylated liposomal doxorubicin, paclitaxel liposome, vincristine sulfate liposome, etc.) have been used in clinical chemotherapy very well [9]. Inorganic materials are also widely used in the preparation of nanomaterials. Mesoporous silica nanoparticles (MSNs) have great advantages in the fields of adsorption, separation, catalysis, and drug delivery [10]. Magnetic Nanoparticles (MNPs), by means of an external magnetic field, can increase the aggregation of MNPs at the tumor site and reduce the distribution in normal tissues. Furthermore, MNPs have the functions of hyperthermia and imaging, and its super paramagnetism makes it an obvious advantage as MRI (magnetic resonance imaging) contrast agent [11]. Other metal nanoparticles, such as gold nanoparticles (GNPs), can inhibit tumor angiogenesis by themselves and have photothermal effects as well [12]. However, these nanomaterials have shown great success in treating tumors and reducing adverse reactions. The presence of TME still bring limitations for nanomedicines to treat tumors. Aiming for the acidic pH, hypoxia and abundant ATP quantity conditions of TME, NPs response to different stimuli were developed, which could remove the obstacle of low accumulation with enhanced permeation and retention (EPR) effect in the tumor [13]. Nano-architectures established by virus-like particles, polymer, inorganics, micelle, self-assembled proteins, liposomes, polypeptides with suitable volume ratio, and tunable morphologies can achieve the purpose of broad spectrum, low toxicity, and low drug resistance. These NPs can not only reach the tumor site precisely, but also load much more lipophilic drug molecules by its special hollow structure, cut off the interaction between tumor cells and the microenvironment, and inhibit the proliferation of tumor cells more efficiently [14,15,16].

The role of the TME during nano-targeted tumor treatment strategies has been reviewed somewhere, however, most research just focused on part of the compositions or stimuli categories [17,18,19]. This review aims to elaborate the components and physiological conditions of TME, summarize the nano-architectures response to physiological barriers or unique constituents, and discuss the prospect of nano therapeutics in TME (Figure 1).

## 2. Special Characteristics of Tumor Microenvironment

### 2.1. Major Constituents of TME

Immune cells, ECM, cancer associated fibroblasts (CAFs), tumor vessels, exosomes and chemokines are vital constituents of TME, all of which participate in tumor progression and invasion particularly (Figure 2). Here, we will concentrate on the principles of their respective activities.

#### 2.1.1. Immune Cells and Chemokines

The functions of immune cells and chemokines are controversial in TME. Current research showed that there were a variety of immune cells in the inflammatory microenvironment, including adaptive immune cells as T lymphocytes (T cells) and B lymphocytes (B cells), innate immune defense cells as macrophages, natural killer (NK) cells, dendritic cells (DCs), and myeloid-derived suppressor cells (MDSCs) [20,21,22]. In the inflammatory microenvironment, the special phenomenon-“immune escape” prevents cancer cells from being recognized by killer cells such as CD8^+^ T cells and NK cells, making it easier for cancer cells to survive, infiltrate, and metastasize [5,23,24]. Among them, tumor associated macrophages (TAMs), regulatory T cells (Tregs) and MDSCs play vital roles in the tumor immunosuppression environment. Recruitment and differentiation of these immunosuppressive cells depend on the presence of numerous cytokines in the microenvironment [25,26,27].

Macrophages can be divided into M1 and M2 type according to different phenotypes and functions. M1 macrophages with tumoricidal effects can inhibit tumor growth and secrete pro-inflammatory cytokines like tumor necrosis factor-α (TNF-α), interleukin (IL)-6, and IL-12 [26,28]. M2 macrophages activated by IL-4, IL-10, and IL-13 hold the function of secreting cytokines such as vascular endothelial growth factor (VEGF), epidermal growth factor (EGF), and transforming growth factor-β (TGF-β). All of them participate in promoting repair, suppressing immune response, and angiogenesis [29,30]. M2 macrophages in the tumor microenvironment account for a much higher proportion than M1 macrophages [26]. Programmed cell death 1 (PD-1) expressed on M2 macrophages combining with the programmed cell death ligand 1 (PD-L1) expressed on tumor cells exerted immunosuppressive effect [31]. Meanwhile, tumor cells highly express CD47, which is the ligand of signaling regulatory protein α (SIRPα), an immune checkpoint found on macrophages [32,33]. Binding of CD47-SIRPα axis will suppress phagocytosis effectively. Therefore, effectively improving phagocytosis of macrophages or transforming M2 macrophages into M1 type is the main therapeutic direction in current research.

Tregs, which are abundant in the tumor stroma is a specific subgroup of CD4^+^ T cells expressing the transcription factor Foxp3, CD25 and cytotoxic T-lymphocyte-associated antigen-4 (CTLA-4) [34,35,36]. The chemokine ligand (CCL) 22 produced by macrophages and tumor cells can bind to chemokine receptor (CCR) 4 expressed on Tregs, consequently recruiting Treg into TME and leading to tumor growth and poor patients’ outcomes with its immunosuppressive function [37]. Tregs create an immunosuppressive environment through the activities of cell surface molecules (Foxp3, CTLA-4, CD25, CD39, CD73, TIGIT), secretion of cytokines (IL-2, IL-10, TGF-β, CCR4) and immune molecules (granzyme, cyclic AMP, and indole-amine-2,3-dioxygenase (IDO)) [36,38,39,40,41,42]. Therefore, blocking the functional molecules expressed on Tregs such as CTLA-4, PD-1, CCR4, TGF-β, Foxp3, or completely eliminating the presence of Tregs can improve the immune escape effect of tumor inflammatory microenvironment.

MDSCs are a group of heterogeneous cells that lack lymphoid markers with multi-directional differentiation potential and immunosuppressive function. This group includes immature DCs, macrophages, granulocytes, and other myeloid cells in the early stage of differentiation [43]. Immune suppression by MDSC involves several complex mechanisms. Due to its suppression of T cells and NK cells in TME, the accumulation of MDSCs is one of the main reasons for tumor immune unresponsiveness [44]. MDSCs can reduce local tryptophan levels due to the activity of IDO to reduce the proliferation of T cells [45]. Peroxynitrite (PNT) produced by MDSCs can alter chemokines and block the entrance of CD8^+^ T cells. MDSC also induce Tregs and affect function of NK cells by producing immunosuppressive cytokines like IL-10 and TGF-β [46]. Besides the influence of differentiation of TAMs, MDSCs also promote angiogenesis by secreting factors compensating for VEGF [27,47]. Immunotherapy targeting MDSCs provides a new therapeutic strategy for anti-tumor therapy.

#### 2.1.2. Extracellular Matrix

ECM contains proteins, glycoproteins, proteoglycans, polysaccharides, and other components. All of them provide structural support for tissue organization and promote information transmission between cells [48]. In normal tissues, connective proteins and adhesive proteins in ECM keep connection between cells and maintain tissue homeostasis [49]. However, in solid tumors, remodeled ECM affects the migration and invasion of cells, and promotes the occurrence and malignant progression of tumors. Working as information transmitter between ECM and other cells, integrins are highly expressed on tumor cells and vascular endothelial cells, and usually affect the function of some immune cells and fibroblasts. Integrins on the surface of tumor cells regulate cell protrusion and adhesion in the process of tumor migration. Meanwhile, they mediate the function of multiple matrix metalloproteinases (MMPs) like MMP2, MMP9, and MMP14 to remodel ECM [50,51]. ECM remodeling is mainly regulated by MMPs, and proteases such as serine acid/cysteine [52]. In addition, the remodeled dense ECM slows the penetration and diffusion of large molecules to create a high-pressure environment, thereby resulting in therapy limitation [53,54].

#### 2.1.3. Cancer Associated Fibroblasts

CAFs are one of the main components of TME. Unlike resting fibroblasts, CAFs metabolize vigorously and secrete large amounts of proteome, including cytokines, chemokines, and various protease CAF spindles [55,56]. CAFs also provide structural support for tissues and act as a transmitter of information between cells [57]. Due to the expression of serine protease, fibrinogen activator, and MMPs on CAFs, ECM is hydrolyzed and reconstructed. In addition, CAFs can also express a variety of cytokines and proteases, such as stromal cell- derived factor 1 (SDF1), VEGF, MMPs, and monocyte chemotactic protein-1 (MCP-1) to promote tumor growth, metastasis, and angiogenesis [58,59,60]. In addition, CAFs support cancer progression through changes of metabolism. In tumors, p38 signal of CAFs activates by cancer cells, the fibroblast-derived p38-regulated cytokines mobilize glycogen in cancer cells, which utilizes by cancer cells for glycolysis, promoting cancer invasion and metastasis [61]. In breast cancer, glycolytic CAFs provide extra pyruvate and lactate for augmentation of mitochondrial activity of tumor cells, which confers tumor cells with multiple drug resistance [62]. CAFs also regulate cancer cell metabolism independently of genetic mutations of cancer cell. FAK-depletion in CAFs promoting chemokin production, enhancing malignant cell glycolysis by activating protein kinase A via CCR1/CCR2 axis [63]. Therefore, NPs targeting CAFs can prevent tumor cells growing in numerous ways.

#### 2.1.4. Exosomes

Exosomes are extracellular vesicles typically ~30 to ~200 nm in diameter and containing genetic material, proteins, and lipids [64]. They act as powerful signaling molecules connecting cancer cells and the surrounding components. Exosomes secreted by tumor cells carry miRNAs to regulate vascular endothelial cells. This phenomenon destroys the barrier of endothelial cells, and then allows cancer cells to enter the blood vessels, promoting tumor spread and metastasis [65]. Exosomes secreted by leukemia cells can promote the activation of CAFs. In breast cancer research, exosomes secreted by CAFs promote the invasion and metastasis of cancer by activating the Wnt-pathway [66]. Astrocytes in the brain metastatic microenvironment secrete exosomes loading miRNAs, which specifically downregulate tumor suppressor gene PTEN and lead to metastatic colonization [67]. Cancer exosomes inhibit the cytotoxic of CD4^+^ T cells, CD8^+^ T cells, and NK cells [68]. Exosomes also inhibit the differentiation of DCs and MDSCs [69,70]. Exosomes derived from cancer cells definitely have short-and long-term effects on cancer progress. Treatments targeting exosomes might be new directions of tumor therapy.

#### 2.1.5. Tumor Vasculature

The regeneration of vasculatures is a very complicated progress in TME. Vascular endothelial cells regulated by the angiogenic factors can affect the migration and proliferation of tumor. The new vessels formed by adhesion of loosely endothelial cells provide chances for tumor growth and distant metastasis [71]. A variety of cells and growth factors are involved in this process, such as vascular cell adhesion molecule-1 (VCAM-1), α(v)β(3) integrin, VEGF, TGF, platelet-derived growth factor (PDGF), and angiogenin. Among them, VCAM-1 and α(v)β(3) integrin not only promote the proliferation and differentiation of endothelial cells but also improve vascular permeability [72,73]. PDGF, angiopoietin, and TGF secreted by tumor cells can also affect the action of peripheral cells, vascular maturation, and integrity. In addition, the highly abnormal and dysfunctional system of tumor blood vessels can also lead to impaired ability of immune effector cells to penetrate solid tumors. Therefore, the normalization of tumor blood vessels can enhance tissue perfusion and improve the infiltration of immune effector cells, thus enhancing therapy effects [74,75].

### 2.2. Physiological Condition of TME for Imaging and Targeting

#### 2.2.1. Hypoxic Condition and Acid Microenvironment

Normal tissue is powered by mitochondrial oxidative decomposition, while cancer cells are mostly powered by glycolysis, a reprogrammed way known as the “Warburg effect” [76]. The majority of tumors are lack of adequate blood supply, and then hypoxic regions appear, where metabolize glucose into lactic acid through anaerobic glycolysis. When a large amount of lactic acid accumulates in the tumor cell, the proton pump transports H+ to the extracellular environment, resulting in an acidic extracellular environment (pH = 5.6–6.8) [77]. During glycolysis, the hypoxia inducible factor (HIF) can regulate glycolysis enzymes (HK1, HK3, TGF-2, et al.) to affect the energy metabolism and proliferation of tumor cells [78,79]. Therefore, utilizing the acid environment of the TME to design a platform acting on HIF-1 could be a new treatment strategy for tumors [80].

#### 2.2.2. Extracellular ATP Content

It is an established notion that extracellular adenosine-5’-triphos-phate (ATP) is one of the major biochemical constituents of TME [81]. Mitochondria are where ATP is produced. It is different between normal cells and cancer cells with mitochondrial metabolism. Although there is little oxygen in cancer cells, in mitochondria, glycolysis is preferential for providing energy, which is called Warburg effect. The reprogramming metabolism in cancer is regulated by central regulators of glycolysis such as HIF-1, Myc, p53, and the PI3K/Akt/mTOR pathways [82]. In tumor tissues, these active pathways promote glycolysis in hypoxia, promoting mitochondria to produce large amounts of ATP. The sharp different concentration of ATP contrast between extracellular (<0.4 mM) and intracellular (1–10 mM) is a characteristic of TME, and the use of ATP can be a practical way for regulating drug release [83,84]. In addition, tumor cells usually metabolize vigorously, and once there is a lack of energy, cell damage happens. The damage of plasma membrane is a recognizable origin of ATP upregulating. Besides the cell injury, the hypoxic-induced stress of TME is also a strong stimulus for ATP release [85]. The ATP concentration in TME is remarkably more than those in normal tissues (10–100 nM) [84]. Based on such a concentration difference, the ATP stimulating response system can be designed to ensure the drugs reach the tumor site more accurately.

#### 2.2.3. Redox Condition

Many organelles, such as cytosol, mitochondrion, and nucleus, contain very high concentrations of glutathione (GSH). In cancer cells, the concentration of GSH is 100–1000 folds of the normal tissue. Due to the existence of thiol groups, GSH can act as electron donors (reducing agents) for developing smart NPs [86].

## 3. Microenvironment-Targeted Nano-Delivery System as a Promising Strategy

The drug therapy for cancer has been unable to exert its maximum effect due to insufficient orientation, pharmacokinetic obstacles. In order to overcome the shortcomings of traditional drug delivery methods, a new method, a nano-delivery system, is being researched. It can bring drugs accurately to the tumor site and prolong the half-life of the drug in vivo. According to different conditions between TME and normal tissue, NPs are designed to be new solutions for tumor imaging and treatment (Figure 3). All the works mentioned were summarized in Table 1 to reflect the latest development of nano-target strategies applied to TME.

### 3.1. Major Composition Mediated Nanoparticles in TME

#### 3.1.1. Nanoparticles Targeting Immune Cells

The immunosuppressive microenvironment is one of the main reasons for the poor antitumor effect in vivo [87]. For M2 macrophages, the most popular methods are reducing M2 macrophages, blocking the immune suppressive function of M2 by blocking PD-1/PD-L1 and the CD47-SIRPα axis [88]. Qian and others designed a fibrin gel capsulate calcium carbonate NPs used in the surgical wound which would polarize M2-like macrophages to a M1-like phenotype. The pre-loaded anti-CD47 antibody in this vector further block the “don’t eat me” signal in cancer cells [89]. Shi et al. utilized precision nanopart icle- based reactive oxygen species photogeneration to reprogram M2 macrophages to M1 macrophages, then recruited cytotoxic lymphocyte (CTL) and direct memory T-cells to make the tumoricidal response more effective [90]. Both gold nanoparticles (AuNPs) and silver nanoparticles (AgNPs) can modulate ROS and reactive nitrogen species (RNS) to activate inflammatory signaling pathways, which can re-polarize macrophages to M1-like phenotypes [91]. We built a multifunctional nanoplatform (FA-CuS/DTX@PEI-PpIX-CpG nanocomposites) for synergistic PDT, PTT, loading DTX to enhance immunotherapy of anti-PD-L1, and polarizing myeloid-derived suppressor cells (MDSCs) toward M1 phenotype successfully in breast cancer (Figure 4) [92].

Recently, interaction between the tumor metabolism and immunity has been proved to be a potential therapeutic strategy. A mannosylated lactoferrin nanoparticulate system (Man-LF NPs) is developed. It facilitated dual-targeting biomimetic codelivery of shikonin and JQ1 to target the macrophage marker mannose receptor and LRP-1. JQ1 itself is a PD-L1 checkpoint blockage that can combine with Man-LF NPs and reduce the generation of immune cells such as Tregs [93]. Macrophages in TME have also contributed to tumor diagnosis and localization, Kim et al. made imaging macrophages in tumors possible towards a pharmacokinetically optimized, ^64^Cu-labeled polyglucose nanoparticle (Macrin) for quantitative positron emission tomography (PET). This technique not only detected the number of macrophages, but also contributed to the effective image of tumor location [94].

In general, the expression of receptor tyrosine kinases (RTKs) in cancer cells can activate the STAT3/5 signaling pathway, which promotes the secretion of TH2 cytokines and then promotes the survival of CD4^+^ Foxp3^+^ Tregs [95,96]. Thus, the sunitinib-targeting receptor tyrosine kinase drug transferred by nanomaterial has been used to decrease CD4^+^ Foxp3^+^ Tregs and MDSCs [97]. Tlyp1 peptide coupled nanoparticles, combined with anti-CTLA4 immuno-checkpoint inhibitors targeting microenvironments, can also enhance imatinib’s ability of decrease Tregs by inhibiting the phosphorylation of STAT3 and STAT5 signaling pathways [98]. A CpG self-crosslinked nanoparticles-loaded IR820-conjugated hydrogel with dual self-fluorescence to exert the combined photothermal-immunotherapy was designed in a melanoma model. These NPs improve the immune response of adjuvant through adjusting the quantity of CD8^+^ T cells, DCs, Tregs, and MDSCs in TME [99].

Stimulator of interferon genes (STING) could enhance tumor immunogenicity, and researchers found, when packing STING into NPs, that its activity to 2’3’ cyclic guanosine monophosphate-adenosine monophosphate (cGAMP) enhanced [100]. IDO,TGF-β, IL-10, and IL-35 also have the abilities to modulate the immune microenvironment [101]. IDO is a rate-limiting enzyme of human tryptophan metabolic that can oxidize tryptophan into canine urine. IDO directly inhibits the function of T cells and enhances the immunosuppressant effect of Tregs, thereby mediating the effect of local immune tolerance and promoting the immune escape of tumors [102,103]. IDO is increasingly incorporated into the nano therapeutics system to regulate the outcome of immunotherapy interventions. In melanoma, Cheng and colleagues built a peptide assembling nanoparticle, which concurrently blockade immune checkpoints and tryptophan metabolism towards on-demand release of a short d-peptide antagonist of programmed cell PD-1 and an inhibitor of IDO [104]. Aimed at TGF-β, Xu et al. silenced TGF-β in microenvironments to solve the problem that the combined vaccine of tumor antigen (Trp 2 peptide) and adjuvant (CpG oligonucleotide) has a poor effect on melanoma [105]. Our team constructed CpG capsuled Cu_9_S_5_@mSiO_2_-PpIX@MnO_2_ NPs to promote infiltration of CTLs in tumor tissue, and further upregulated interferon gamma (IFN-γ) to promote immune response [106].

Suppressing the function of immune-tolerant cells and promoting the anti-tumor effect of cells have been the key points of tumor immunotherapy. Regulation of immune tolerance in TME combined with nanomaterials can effectively avoid the obstacles caused by microenvironments for drug entry. Therefore, regulating the function of immune cells in microenvironments is one of the key tasks of the nanomaterials system.

#### 3.1.2. Nanoparticles Targeting CAF

CAF is the main source of growth factors, chemokines, ECM proteins, and matrix degrading enzymes in TME. It can produce a variety of growth factors and cytokines to promote the survival and invasion of tumor cells [107]. Kovacs and colleagues found that gold-core silver-shell hybrid nanomaterials could reduce the tumor promotion by attenuating behavior of CAFs [108].

According to the regulation of CAFs of immune cells, Hou et al. developed a nanoemulsion (NE) formulation to deliver fraxinellone (Frax). This NP was around 145 nm length, could be taken by CAFs efficiently, and accumulated in the TME. Combining with a tumor-specific peptide vaccine will enhance tumor-specific T-cell infiltration and activate death receptors on the tumor cell surface (Figure 5) [109]. Recently, NPs targeting CAFs for tumor therapy mainly focused on destroying the tumor tissue to promote drug penetration and reprogramming immune TME. Since CAFs participate in cancer glucose metabolism immediately, NPs targeting CAFs about decreasing glycolysis of cancer cells remain to be developed.

#### 3.1.3. Nanoparticles Targeting ECM

People try to find ways to destruct the structure of ECM. As the most well-known matrix enzyme, MMPs, particularly MMP2 and MMP9, were frequently applied in NPs [110,111]. Many systems that respond to enzymes such as MMPs are also used in drug delivery and imaging. Ji et al. utilized pirfenidone (PFD) loaded MRPL (MRPL-PFD), a MMP2 responsive peptide-hybrid liposome, to downregulate the components of ECM, thus increasing the penetration of drugs in pancreatic cancer tissue (Figure 6) [112]. Other enzymes acting on the microenvironment can also degrade the structure of ECM. Hyaluronidase (HAase) can break down hyaluronan and then enhance the efficacy of nanoparticle-based PDT. Utilizing HAase and DOX together will also increase cancer mortality [113,114,115]. Blocking collagen and integrin signaling for anti-fibrotic therapeutic strategy can also be considered in future treatment for ECM to improve drug delivery [116].

#### 3.1.4. Nanoparticles Targeting Exosomes

As a natural intercellular shuttle of miRNA, exosomes affect a series of physiological and pathological processes in receptor cells or tissues, and are ideal nano carriers for nucleic acid targeted delivery in vivo [117]. The antigen presenting function of dendritic cells was utilized to develop a single membrane vesicle-based vaccine, which would participate in repressing both melanoma (B16) and Lewis lung carcinoma (LLC) tumor growth [118]. Utilizing exosomes to load NPs showed perfect biocompatibility. Xiong and co-workers built NPs together laurate-functionalized Pt (IV) prodrug (Pt(lau)) and human serum albumin (HSA) with lecithin, capsuled by the exosomes, had a good platinum chemotherapy efficiency (Figure 7) [119].

Exosomes shed by cancer cells have also been designed on cancer diagnosis. Liu et al. made exosomes immobilize on magnetic microbeads to produce fluorescent signal. They qualified the exosomes in plasma samples from breast cancer patients for early diagnosis of cancer in vitro [120]. Lewis et al. built an in vitro probe screening of bio-membrane chips, which was composed of the captured exosomes and other extracellular vesicles in the plasma, and tested the express of glypican-1 and CD63 to diagnose pancreatic ductal adenocarcinom in vitro [121].

#### 3.1.5. Nanoparticles Targeting Tumor Vasculature

Using EPR more effectively and enhancing the permeability of vascular have been widely studied in the latest nano therapeutics [122,123]. There are some reports that used NO to improve the EPR effect in pancreatic cancer and other diseases with low vascular permeability [122,124,125]. Other delivery systems like actively targeting VEGF and α(v)β(3) integrin were also used widely [126]. Integrins play important roles in cell adhesion and cell signaling, and α(v)β(3) integrin is one type of them that can modulate angiogenic endothelial cells. Graf et al. described a NP using cyclic pentapeptide c(RGDfK) to active target α(v)β(3) integrin on cancer cells and tumor neovasculature [127].

To improve the diagnosis of tumor, Youbin and co-workers proposed the poly(acrylicacid) (PAA)-modified NaLnF_4_:40Gd/20Yb/2Er nanorods ((Ln = Y, Yb, Lu, PAA-Ln-NRs) to enhance the shifting of NIR-IIb (a general in vivo fluorescence imaging technology), which successfully imaged the vessels of small tumors (about 4 mm), metastatic tissue (about 3 mm), and even brain vasculatures (Figure 8) [128]. Cecchini et al. reported a nanoMIPs against VEGF coupled with quantum dots (QDs) for tumor imaging in melanoma [129]. In cholangiocarcinoma, α(v)β(3) integrin also combined with aggregation induced emission (AIE) for image-guided PDT, and presented a good antitumor response [130].

#### 3.1.6. Nanoparticles Targeting Multiconstituents

In addition to targeting single constituents, some NPs can regulate multiple barriers and have also been simultaneously designed for tumor therapies. Targeting tumor cells and immune cells simultaneously can effectively reduce the immune escape phenomenon of TME. Shi et al. designed a versatile calcium ion nanogenerator. The degradation and release of Ca^2 +^ by nanoparticles can promote the maturation of DCs by promoting autophagy of DCs, and it can promote tumor cells to produce damage-associated molecular patterns (DAMPs), further maturing DCs and the enhanced infiltration of CTLs at the same time [131]. In addition, researchers have combined exosomes with immunotherapy. Xie’s team synthesized responsive exosome nano-bioconjugates. They modified exosomes derived from M1 macrophages with antibodies of CD47 and SIRPα. The broken benzoic-imine bonds are cleaved to release antibodies of SIRPα and CD47 in the acidic TME abolished the “don’t eat me” signal between tumor cell and macrophages [132]. The native M1 exosomes reprogram the M2 macrophages to M1 effectively at the same time [132].

### 3.2. Physiological Condition Mediated Nanoparticles in TME

#### 3.2.1. Hypoxic Stimulus

The hypoxic condition in tumor microenvironment is considered to play an important role in malignancy and progression of cancer. Hypoxic areas of tumors also bring obstacles to radiation therapy due to the oxygen free radicals [133]. Utilizing this characteristic, the low oxygen response nano-delivery systems were produced.

The approaches developed thus far can be classified into three categories: countering hypoxia, exploiting hypoxia, and disregarding hypoxia [134]. (i) Directly or indirectly elevating O_2_ concentration to counter hypoxia is a promising way to improve the efficiency of tumor therapy, especially for photodynamic therapy (PDT) and radiation therapy (RT). PDT generates reactive oxygen species (ROS) by using light-excited photosensitizer (PS), resulting in cell apoptosis and microvascular damage [125]. Red blood cells (RBCs) carrying hemoglobin molecules are primary oxygen sources in our body. Because the efficacy of PDT is deeply oxygen-dependent, a technique named RBC-facilitated PDT was developed to improve hypoxia conditions in tumors. Wei et al. showed a nanocapsule that encapsulated photosensitizers and tethered the conjugates onto RBC surface. By using biotin-neutravidin-mediated coupling, they conjugated ZnF16Pc (photosensitizer)-loaded ferritins onto each RBC [135]. This new structure, which could overcome low oxygen conditions, showed efficient ^1^O_2_ production to overcome low oxygen conditions and enhanced PDT capacity [135]. For RT, an artificial nanoscale RBC will remarkably enhance the treatment efficacy as well. For example, an artificial blood substitute perfluorocarbon (PFC) was encapsulated with biocompatible poly(d,l-lactide-co-glycolide) (PLGA) and then further coated with a red-blood-cell membrane (RBCM), showing efficient loading of oxygen and significantly enhanced treatment efficiency during RT [136]. With increasing production of H_2_O_2_ in cancer cells, NPs converting endogenous H_2_O_2_ to toxic ROS and decomposing endogenous H_2_O_2_ to O_2_ were rapidly developed. Noble metal nanoparticles like Mn, Au, Pt, and Ir are well known for their catalytic performances in various fields [137]. MnO_2_ is a common material to enhance PDT treatment and imaging. Mn^2+^ could react with H_2_O_2_ in the tumor, then downregulate the expression of HIF-1 to increase oxygen content and optimize MRI imaging [138,139]. A nanoplatform based on mesoporous polydopamine (MPDA) modified with Pt also produces O_2_ by decomposing overexpressed H_2_O_2_ in the tumor. Meanwhile, the existence of Pt can act as a nano-factory to provide support for PDT (Figure 9) [140]. (ii) Taking advantage of the deficiency of oxygen molecules is a new approach for drug release and PDT. Yin et al. developed a novel amphiphilic block copolymer radiosensitizers. After optimizing the ratios of carboxyl and metronidazole (MN) groups, PEG-b-P(LG-g-MN) micelles could be used to encapsulate doxorubicin (DOX@HMs) efficiently [141]. Hypoxia-responsive structural transformation of MN into hydrophilic aminoimidazole triggers fast DOX release from DOX@HMs, which acted as high-efficiency radiosensitizers and hypoxia-responsive DOX nanocarriers [141]. Some drugs that are selectively toxic to hypoxic cells like Tirapazamine (TPZ) were designed to combine with oxygen-dependent PDT to enhance bioreductive therapy. Shao’s group developed a core–shell upconversion nanoparticle@porphyrinic MOFs (UCSs) for combinational therapy against hypoxic tumors [142]. TPZ was encapsulated in nanopores of the MOF shell of the heterostructures (TPZ/UCSs), which enables the near-infrared light-triggered production of cytotoxic reactive oxygen species [142]. Furthermore, with the combination of PD-L1, this nanoplatform recruited specific tumor infiltration of cytotoxic T cells and inhibited the metastasis of the tumor as well. Other methods like eliminating the oxygen in the tumor, inhibiting the growth of tumor vessels, and stopping the nutrient delivery to starve the tumor cells still have many challenges [143,144]. (iii) Using new anticancer modalities to disregard hypoxia conditions becomes another innovative antitumor strategy. PDT with diminished O_2_ dependence will effectively overcome its strong oxygen dependence and limitation of treating deep tumors. It has been reported that fractional light delivery may be a superior way to enhance PDT effects due to the reduction of short-term oxygen consumption during PDT [145]. Since the generation of oxygen-irrelevant free radicals is oxygen-independent, and the exploration of UCNP is an inner light source to activate most organic photosensitizers (PSs) to create cytotoxic ^1^O_2_, researchers discovered that the Ru complex displayed excellent type I PDT activity [142,146]. Due to its special Fenton reaction, Fe nanoparticles can produce reactive •OH species with endogenous H_2_O_2_ (Fe^2+^ + H_2_O_2_ → Fe^3+^ + •OH ^+^ OH^−)^ and produce cytotoxic effects without external energy through chemotherapeutic therapy (CDT). Yu et al. fabricated a core–shell structured iron-based NPs (Fe_5_C_2_@Fe_3_O_4_) to release ferrous ions in acidic environments to disproportionate H_2_O_2_ into •OH radicals, and its high magnetization is favorable for both magnetic targeting and T2-weighted MRI [147]. In addition, gold nanospheres, graphene oxide, polydopamine (PDA), and other materials have been widely used as PTT reagents and nano carriers to deliver PDT reagents, so as to overcome the therapeutic limitations of PDT [148,149,150].

#### 3.2.2. pH Response

pH responsive nano-vectors are one of the typical carriers for TME. Chemical bond response to pH is one of the most widely used strategies in pH responsive nano delivery systems. The most common pH sensitive bonds include hydrazone bond, imine bond, oxime bond, amide bond, benzoic-imine bond, orthoesters, polyacetals, and ketals [151,152]. These chemical bonds break in acidic environments to degrade the carrier, and then increase the uptake of tumor cells or accelerate drug release. When it comes to the design of pH-sensitive materials, besides pH sensitive chemical bonds, other main strategies are conformational change, protonation, and charge reversal with pH change [151]. For example, Chen’s team developed a DNA-based stimulus-responsive drug delivery system precisely responding to pH variations in the range of 5.0–7.0. On the face of the gold nanoparticles, one DNA strand was an acti-MUC1 aptamer targeting tumor membrane, the other DNA strand was switchable DNA, which has a linear conformation under neutral or alkaline conditions and self-folds into a triplex under acidic conditions [153].

In cells, nano-switches can react with endosomes and lysosomes and switch to triplex in lysosomes, so as to achieve the goal of accurately drug release (Figure 10) [153]. Nanocarriers involved in protonation/deprotonation are mainly nano liposomes, peptides, and polymers. The phospholipid components in liposomes are usually destabilized under acidic conditions, so as to deliver the contents of liposomes to cells. In mouse cancer models, Guangna et al. combined platelet membrane with the functionalized synthetic liposome; because of its camouflage based on the platelet membrane, this platform enhanced tumor affinity and released DOX in acidic microenvironment more selectively and efficiently [154]. The anionic/cationic polymers with different groups deform various nanocarriers through the change of their hydrophilicity, which lead to drug release [155]. Inorganic salts such as MnO2, CaP, and CaCO_3_ are widely used for pH response NPs because of their acid solubility [156,157,158]. Ma et al. designed a pH-sensitive dye linked peptide substrate of MMP-9 with Fe_3_O_4_ nanoparticles, establishing a Forster resonance energy transfer (FRET) system to detect the invasion and metastasis of tumor by detecting the overexpression of MMP9 [159]. A pH responsive magnetic nanoparticle can combine magnetic hyperthermia with drug delivery dependent on magnetic stimulation, achieving the purpose of targeting TME and tumor treatment at the same time [160].

#### 3.2.3. ATP Response

Since the concentration of ATP in tumor cells is much higher than extracellular environments and ATP is involved in many biochemical reactions in cells, NPs response to ATP were widely developed. ATP sensitive NPs can release drugs without the help of external forces. Zhenqi et al. developed nano ZIF-90 self-assembled from zinc ions and imidazole-2-carboxaldehyde (2-ICA) to deliver DOX. Because the coordination between ATP and Zn^2+^ is much stronger than that between imidazole and Zn^2+^, nano ZIF-90 can be decomposed and respond to ATP [161]. Graphene oxide (GO) has been shown to bind single stranded DNA. When the template DNA contains ATP binding domain and reached ATP specific recognition, it could be circularized upon proximity ligation after hybridizing to linker DNA on the surface of GO [162]. Then, rolling circle amplification was initiated from the 3′-end of the template DNA, and the elongated sequence was hybridized with thousands of signal DNA (conjugated with Cy3), so as to amplify the template DNA, generate fluorescent signals, and achieve the purpose of tumor monitoring [162]. Yuan et al. exploited a ATP binding natural protein, GroEL (a bacterial chaperonin) loading DOX, once in the presence of a critical concentration of ATP in tumor site, it releases drugs [163]. In addition to using ATP as a switch for drug release, another way in which ATP participates is to regulate its expression in cells with nanosystems. Xiao’s team exploited a multifunctional theranostic platform combing CDT with limotherapy. While enhancing the CDT effect to induce apoptosis of cancer cells, nano Se and Mn^2+^ ions inhibited the production of ATP, which made cancer cells starve and further killed tumor cells, monitoring the treatment of tumors by MRI simultaneously [164].

#### 3.2.4. Reduction Response

In addition to GSH, tumor cells also contain thioprotein, Fe^2+^, cysteine, and other reductive substances, and the difference of GSH concentration between tumor cells and TME makes a reduction-responsive drug carrier come true [165,166]. GSH/glutathione disulfide (GSSG) is one of the major redox couples in cells, and adding disulfide bonds to drug carriers is one of the most commonly used methods to build GSH responsive drug carriers. There are many forms of GSH-responsive nano-vehicles (like micelles, nanogels, nanoparticles), so as to improve the drug release successfully. For example, in order to solve the problems of drug resistance caused by cancer stem cells (CSC), Rubone (RUB, a miR-34 activator for targeting CSCs) and DTX were utilized to treat taxane resistant prostate patients. A self-assembled DTX/p-RUB micelles showed good stability in vitro and could be accurately delivered to tumor cells though the EPR effect. After the tumor cells endocytosed the micelles, the micelles expanded and disintegrated due to the protonation of diisopropylaminoethanol (DIPAE) and GSH induced disulfide bond cleavage of acid endocytosis vesicles, which led to the rapid release of DTX and RUB. The release of RUB upregulated miR-34a and regulated the expression of chemoresistance related proteins, thus making tumor cells sensitive to DTX, significantly inhibiting the progress of drug resistance [167]. Ling and colleagues constructed a self-assembled NP platform composed of amphiphilic lipid polyethylene glycol (PEG), and it can effectively deliver Pt (IV) precursor drugs through the elimination of GSH [168].

Some metal oxides like MnO_2_ also have the potential of GSH response. MnO_2_ reacts with GSH in cells to form glutathione disulfide and Mn ^2+^, which leads to the consumption of GSH and enhancement of CDT. In addition to the MRI features of Mn^2+^, MRI monitored chemo- chemical combination therapy is realized [167].

#### 3.2.5. Enzyme Response

Many enzymes like MMPs regulate the function of cellular components and take part in tumor progression. From this prospect, the presence of these abnormal enzymes gives the chance for researchers to build a sensitive system for drug release. The presence of NO can activate endogenous MMP1 and MMP2, and researchers have developed an MSN loaded with a doxorubicin (DOX) and NO donor to enhance the antitumor effect [169]. In addition, PLGLAG peptide and gelatin are both main target proteins of MMP9 and MMP2, which can be widely used to MMP responsive NPs [170,171,172].

According to the Warburg effect, the proliferation of tumor cells mainly depends on aerobic glycolysis, so tumor cells are more sensitive to the change of glucose concentration. Glucose oxidase (GOx), an endogenous oxidoreductase, reacts with glucose and O2 in cells then produce gluconic acid and H_2_O_2_, which can inhibit the proliferation of cancer cells through starvation therapy. In addition, H_2_O_2_ can be transformed into · OH free radical to kill cancer cells and enhance the oxidative stress response of cancer cells [173]. Through GOx, starvation therapy can together with chemotherapy, CDT, PDT, or immunotherapy to explore new strategies for cancer treatment. For example, Mengyu et al. constructed a multifunctional cascade bioreactor based on hollow mesoporous Cu2MoS4 (CMS) loaded with GOx for synergetic cancer therapy by CDT/starvation therapy/phototherapy/immunotherapy [174]. First of all, CMS containing multivalent elements (Cu^1+^/^2+^, Mo^4+^/^6+^) showed Fenton like activity, which could produce · OH and reduce GSH, thus reducing the antioxidant capacity of tumor. Secondly, in hypoxic TME conditions, hydrogen peroxide like CMS can react with H_2_O_2_ to generate O_2_, activate the effect of GOx, start starvation therapy, and regenerate H_2_O_2_. Finally, the regenerated H_2_O_2_ can participate in Fenton like reactions to realize GOx-catalysis-enhanced CDT. At the same time, because of the excellent photothermal conversion efficiency under 1064 nm laser irradiation, CMS killed tumor cells significantly in PDT. More importantly, the PEGylated CMS@GOx-based synergistic therapy combined with anti-CTLA4 can stimulate a robust immune response [174].

#### 3.2.6. Multiply Response

A mesoporous silica-coated gold cube-in-cubes core/shell nanocomposites loading DOX was combined with ArgGlyAsp (RGD) peptide to achieve a platform that can deliver drugs and produce O_2_ in situ. This nano platform simultaneously enhanced photodynamic efficacy, achieving heat- and pH-sensitive drug release and location imaging (Figure 11) [175]. Lan et al. decorated an emerging class of highly tunable two-dimensional material: cationic Hf12-Ru nanoscale metal-organic frameworks (Hf12-Ru nMOF), then functionalized with pH-sensitive fluorescein isothiocyanate and targeting mitochondria, utilized the pH and quantities of O_2_ in the mitochondria to image living cells [176]. Yi and colleagues developed a redox/ATP switchable theranostic NPs. They conjugated a fluorescent probe (FAM) and a quencher (BHQ-1) to ATP, complexed with a GSH-sensitive cationic polymer. This smart NPs loading fluorescent probes can monitor drug release in vivo [177].

## 4. Conclusions and Outlook

Traditional cancer diagnosis, chemotherapy, radiotherapy, surgery, and other treatments have kept the high mortality rate of cancer patients, and this led us to develop new strategies with more accurate diagnoses and more effective treatments. Using NPs to treat cancers is an emerging approach. In addition to targeting tumors themselves, utilizing TME and physicochemical properties to treat and orientate tumors have been a great inspiration and challenge for the development of nanoparticles. More evidence is needed for the clinical application of NPs, and here we summarize current results and several challenges of NPs.

The special conditions of the tumor microenvironment give us superior delivery conditions. The immunosuppressed environment of tumor causes “immune escape” and serves as a good direction for the treatment of tumors. The extracellular matrix, enzymes, and inflammatory factors also provide promising therapeutic targets. The physiological features such as hypoxia, acidic microenvironment, and abundant angiogenesis also give NPs good access conditions to reach tumor site and release drugs.

How to use the particular microenvironment of tumor to design delivered nanoparticles is a big hurdle. A question that remains to be solved is how to deliver drugs to tumor tissues more efficiently and specifically. PEG and zwitterionic materials can effectively reduce the blood clearance rate. In order to improve the biocompatibility, it is also a breakthrough for people to use the biological membranes to cover the material. According to the EPR effect and the abundance of blood vessels in tumors, drugs will be delivered to tumor tissue precisely, hence improving therapeutic efficiency for tumors. However, there is a huge difference between the internal environment of human beings and that of animal models, and how to reduce the side effects of materials and systems is what we need to work on. Clinical trials on nanoparticles are yet to be developed, and we should make more efforts to develop safe and efficient therapy strategies.

## Figures and Tables

**Figure 1 molecules-26-02703-f001:**
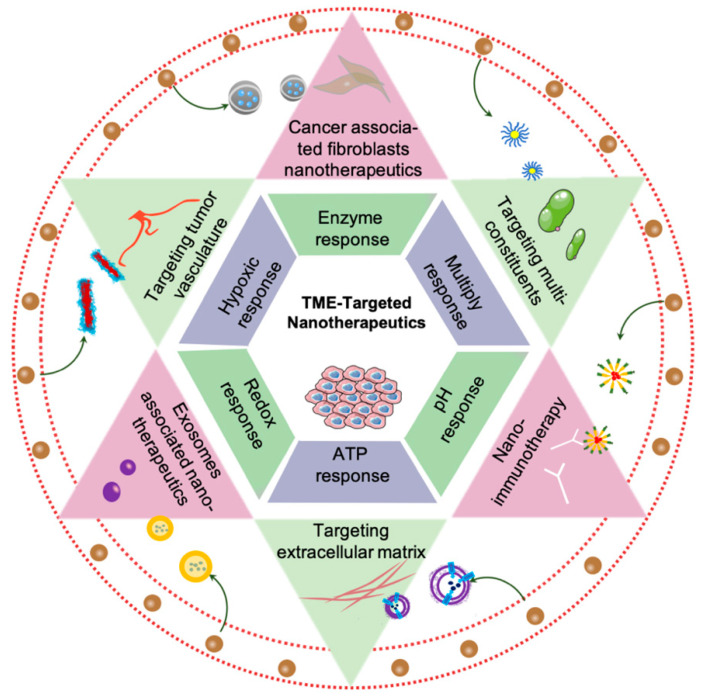
Schematic of nano therapeutics target tumor microenvironment.

**Figure 2 molecules-26-02703-f002:**
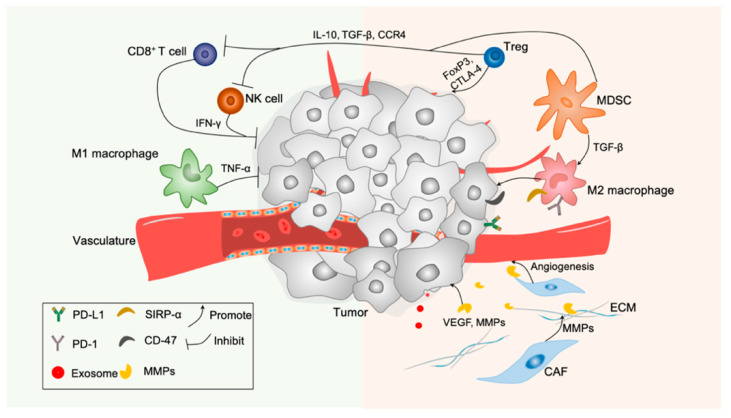
The main components of the tumor microenvironment, including tumor cell, immune cells (such as B cells, T cells, DC cells, NK cells, M1/M2 macrophages), tumor vasculature, ECM, fibroblast, exosomes and cytokines.

**Figure 3 molecules-26-02703-f003:**
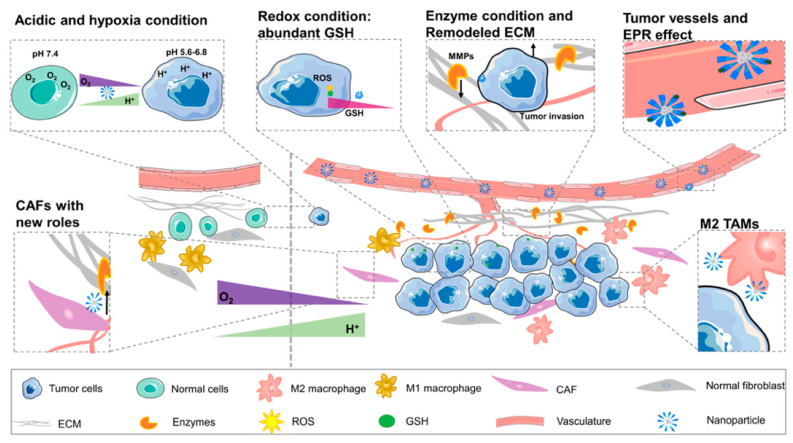
The difference between tumor microenvironment and normal cellular environment (N: normal cellular environment, T: tumor microenvironment). The picture shows new curved blood vessels in TME and EPR effect. Redox condition created by the high content of GSH in tumor cells. Compared with the normal tissue, TME contains a large number of enzymes, anoxic and acidic environments. The percentage of M2 TAM in tumor tissue is larger than that in normal tissue. Special ECM and CAF cells contact with vasculature provides a good condition for angiogenesis and tumor progression.

**Figure 4 molecules-26-02703-f004:**
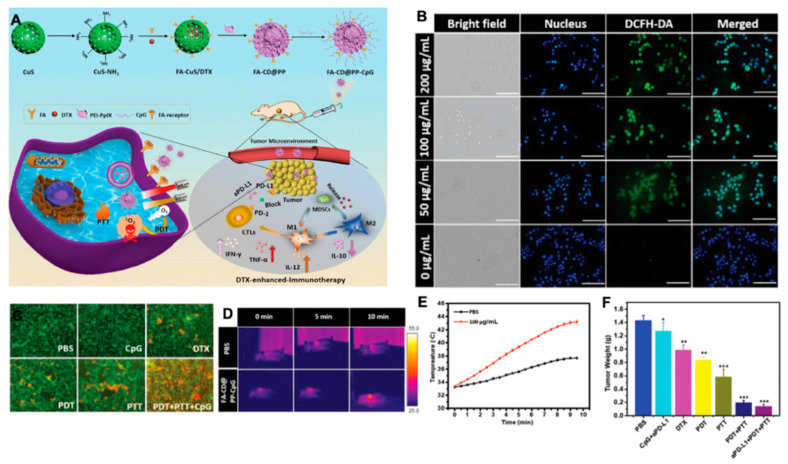
(**A**) Rational design and synthesis, its application in cancer treatment (left), and illustration of FA-CD@PP-CpG for docetaxel-enhanced immunotherapy (right); (**B**) intracellular ROS detection in 4T1 cells incubated with various concentrations of FA-CD@PP-CpG under 650 nm irradiation; (**C**) corresponding fluorescence images of 4T1 cells constrained with calcein AM (live cells, green) and propidium iodide (dead cells, red) after being treated with different conditions. (**D**) The in vivo thermal images of the mice after intravenous injection of PBS and FA-CD@PP-CpG under 808 nm irradiation; (**E**) temperature change curve of tumor sites as a function of irradiation time; (**F**) the weight of tumor tissue in different groups obtained on day 14, adapted with permission from [92].

**Figure 5 molecules-26-02703-f005:**
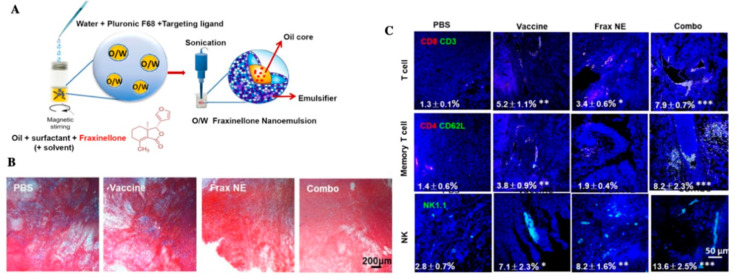
(**A**) Scheme depicting the preparation process of Frax NE; (**B**) Masson’s trichrome stain for collagen; (**C**) confocal and flow cytometric analysis of immune cells infiltration in the TME, adapted with permission from [109].

**Figure 6 molecules-26-02703-f006:**
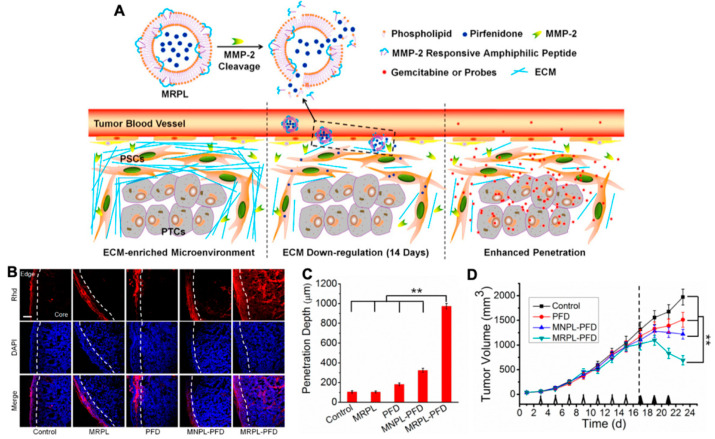
(**A**) Mechanism of MMP-2 Responsive Peptide Hybrid Liposome (MRPL) for downregulation of ECM in pancreatic tumors; (**B**) Rhd penetration and distribution in pancreatic tumor (PSCs/Mia-paca-2 co-implanted) tissues after 2 weeks’ treatment of the different PFD formulations; (**C**) quantification of the depth of Rhd penetration in tumors treated by the PFD formulations; (**D**) the growth curves of PSCs and Mia-paca-2 co-implanted pancreatic tumors in mice treated by the different PFD formulations, adapted with permission from [112].

**Figure 7 molecules-26-02703-f007:**
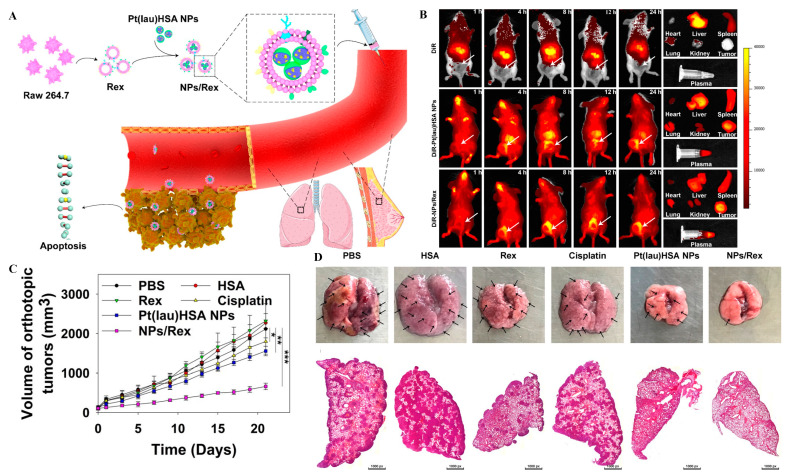
(**A**) Schematic illustration of the Pt(lau)HSA NP-loaded exosome platform (NPs/Rex) for efficient chemotherapy of breast cancer; (**B**) biodistribution of DiR, DiR-Pt(lau)HSA NPs, and DiR-NPs/Rex in 4T1 tumor-bearing BALB/c mice; (**C**) the volume of orthotopic tumors; (**D**) typical lung tissues with visualized metastatic nodules (black arrows) and H&E for metastatic nodules of lungs in each group, adapted with permission from [119].

**Figure 8 molecules-26-02703-f008:**
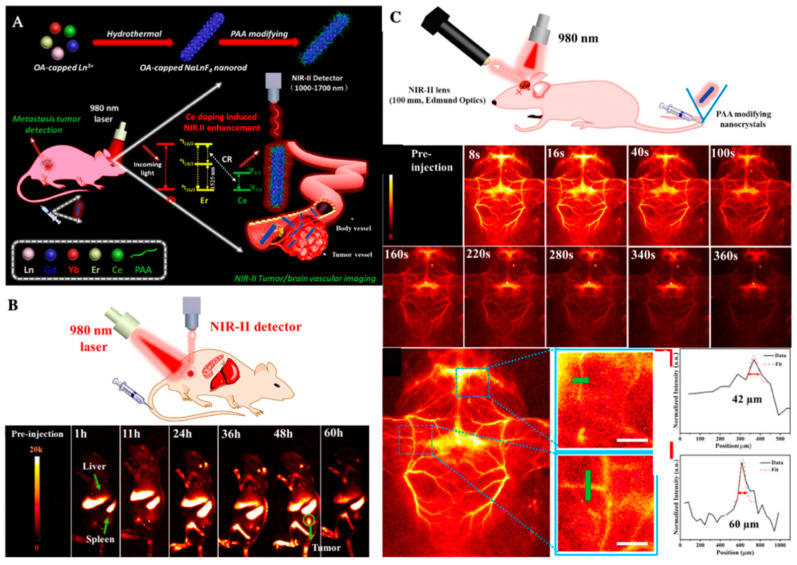
(**A**) Schematic illustration of the enhanced NIR-IIb emission of PAA-Ln-NRs via Ce^3+^ doping for non-invasive tumor metastasis/vascular visualization and brain vessel imaging; (**B**) schematic illustration of in vivo small tumor diagnosis by using PAA-Lu-NRs, and the NIR-IIb bioimaging of tumor-bearing mouse after intravenously injecting PAA-Lu-NRs at different time periods; (**C**) schematic illustration of in vivo noninvasive brain vessel imaging by using the in vivo imaging system, fast brain vascular imaging of a mouse with hair removed and cross-sectional fluorescence intensity profiles along the green lines of the mouse, adapted with permission from [128].

**Figure 9 molecules-26-02703-f009:**
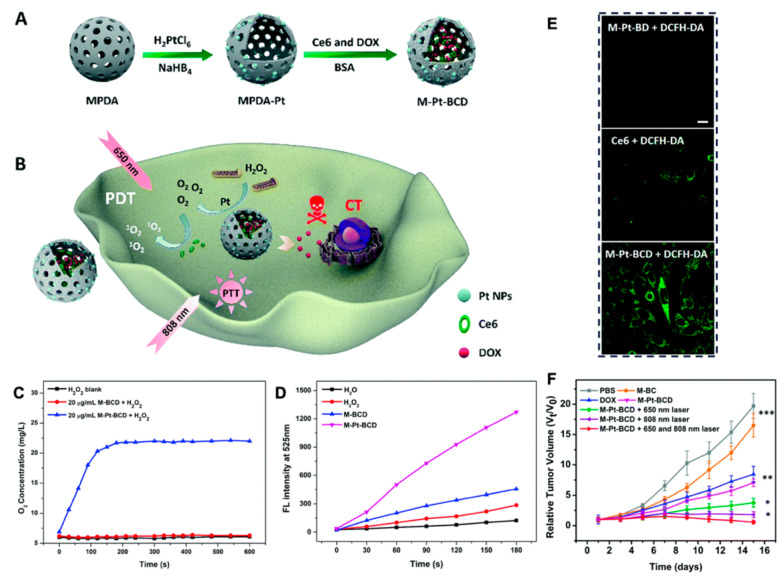
(**A**) Schematic illustration of the synthesis of MPDA-Pt-BSA/ Ce6/DOX (M-Pt-BCD); (**B**) schematic illustration of the application of M-Pt-BCD for enhanced- PDT and synergistic therapy; (**C**) O_2_ generation of H_2_O_2_ blank, M-BCD and M-Pt-BCD; (**D**) ^1^O_2_ production efficiency of H_2_O, H_2_O_2_, M-BCD and M-Pt-BCD; (**E**) confocal microscopic images of cellular ^1^O_2_ levels detected by DCFH-DA staining upon 650 nm irradiation; (**F**) tumor growth curves, reproduced by permission of The Royal Society of Chemistry [140].

**Figure 10 molecules-26-02703-f010:**
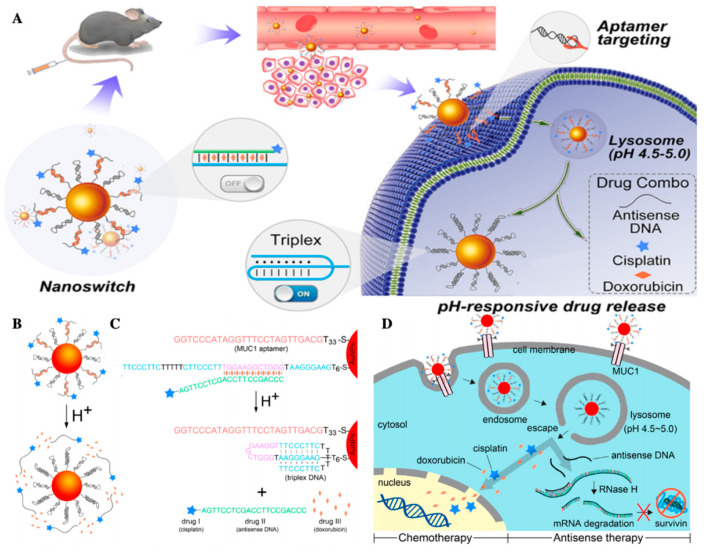
(**A**) Schematic illustration of the DNA-based stimulus-responsive drug delivery system; (**B**) pH-responsive regulation of the nanoswitch and drug release; (**C**) schematic illustration of DNA strands and the multidrug loaded on the surface of AuNPs; (**D**) intracellular pH-responsive multidrug delivery and release, adapted with permission from [153].

**Figure 11 molecules-26-02703-f011:**
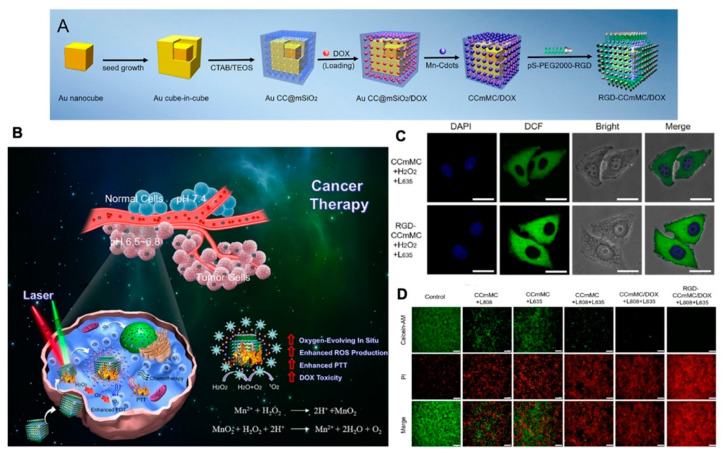
(**A**) Schematic illustration of the synthesis process for the versatile RGD-CCmMC/DOX nanovehicles; (**B**) schematic illustration of the therapeutic mechanism of the RGD-CCmMC/DOX nanoplatforms to enhance the overall anticancer efficiency of triple-combination photodynamic/photothermal/chemo-therapy in a solid tumor; (**C**) CLSM images of 4T1 cells treated with different formulations under laser irradiation. The production of intracellular ROS and O_2_ generation were measured by the green fluorescence intensity of DCF; (**D**) fluorescence microscopy images of 4T1 cells with various treatments using Calcein AM/PI staining, adapted with permission from [175].

**Table 1 molecules-26-02703-t001:** Nanoparticle approaches for targeting TME in this review.

Target	Loading Drugs	Nanocomposites and Outcomes	Animal Models/Cell Lines	Ref.
1. Physiological condition				
Hypoxia response	-	P-FRT-RBCs to enhance PDT, show efficient ^1^O_2_ production	U87MG-bearing subcutaneous models	[135]
	-	PFC@PLGA-RBCM NPs to enhance RT	4T1 tumor-bearing mice model	[136]
	acriflavine	ROS responsive ACF@MnO_2_ NPs to guide RT and MRI	CT26 -bearing mice model	[138]
	DOX	MPDA-Pt-BSA/Ce6/DOX combined PDT and PTT	MDA-MB-231 tumor -bearing mice model	[140]
	DOX	DOX@HMs to enhance chemoradiotherapy	4T1 tumor-bearing mice model	[141]
	TPZ and a-PD-L1	TPZ/UCSs combined with PDT to activate chemotherapy and immunotherapy	CT26-bearing mice model	[142]
	coumarin	coumarin-modified cyclometalated Ru (II) complexes for better PDT effect	HeLa cell-bearing mice model	[146]
	-	PEG/Fe_5_C_2_@Fe_3_O_4_ NPs with magnetic targeting for produce reactive •OH species and MRI imaging	4T1 tumor-bearing mice model	[147]
pH response	DOX, cisplatin, and asDNA	AuNPs with DNA bands, release three drugs due to nanoswitch changes conformation in acidic condition	MCF-7, Hela, L02 cells/Balb/c nude mice model	[153]
	DOX	pH-sensitive PEOz-liposome-dox NPs for drug delivery	CT26 and 4T1 -bearing mice model	[154]
	-	ANNA/ MMP-9/PEGylated Fe_3_O_4_ particle for MR imaging to guide tumor invasion in vivo	BALB/c nude mice bearing human colon cancer	[159]
ATP response	DOX	AP-ZIF-90@DOX, dual responsive to high ATP and low pH condition to release DOX in tumor cells	MDA-MB-231 tumor bearing mouse	[161]
	DOX	DNA/MSN/FA/DOX NPs for drug release and fluorescence imaging	HeLa cells	[162]
	-	MCDION-Se with CDT, inhibit the generation of ATP, thus starving cancer cells	HeLa and HK-2 cells/BALB/C nude mice model	[164]
Reduction response	DTX and RUB	DTX/p-RUB micelles, regulate the expression of chemoresistance	DU145-TXR and PC3-TXR cells/mice model	[167]
	Pt prodrug	self-assembled PEG/Pt (IV) NP through GSH-exhausting effect to delivery safer and more effective	A2780cis tumor-bearing athymic nude mice	[168]
Enzyme response	DOX	MSN loaded with DOX and NO donor (S-nitrosothiol) to create DN@MSN, activate MMP and degrade collagen in the tumor ECM	4T1 tumor-bearing mice model	[169]
	anti-CTLA4	PEGylated Cu_2_MoS_4_ (CMS)@GOx, promote CDT, PDT, PTT and starvation therapy	HeLa cell-bearing mice model	[174]
Multiply response	DOX	RGD-CCmMC/DOX nanovehicles achieve heat- and pH-sensitive drug release with precise control to specific tumor site	4T1 tumor-bearing mice model	[175]
	-	Hf12-Ru nMOF for ratiometric pH and oxygen sensing in mitochondria for monitoring pH and O_2_ in live cells	CT26 cell line	[154]
	DOX	FAM-ATP/BHQ-1-cDNA@DOX NPs,can monitor drug release in vivo	HeLa, HepG2, or MCF-7 cell line	[177]
2. Immune cells				
Macrophages	anti-CD47 antibody	aCD47@CaCO_3_ nanoparticles encapsulated in fibrin gel to scavenge H^+^ in the surgical wound, polarize TAM	Female C57BL/6 mice; B16F10 cell line	[89]
	-	MAN-PLGA and MAN-PLGA-N NPs affected by acidic pH, disrupt the endosome/lysosome membrane help rise ROS and M1 macrophages	BALB/C mice; 4T1, B16, RAW264.7 cell lines	[90]
	-	AuNPs and AgNPs modulate the reactive ROS and RNS production, downregulate TNF-α and IL-10	murine fibrosarcoma model	[91]
	DTX	FA-CuS/DTX@PEI-PpIX- CpG nanocomposites cooperated with PDT and PTT, enhance immunotherapy successfully	4T1-tumor-bearing mice model	[92]
	shikonin and JQ1	Man-LF NPs targeting mannose receptor and LRP-1 expressed on cancer cells and TAMs, inducing immune cell death, repressing glucose metabolism and repolarizing TAMs	CT26-tumor-bearing mice model	[93]
	-	^64^Cu-labeled polyglucose nanoparticle (Macrin) for PET can image the macrophages in tumor, to evaluate TAM-targeted therapy	KP-tumor bearing lungs-C57BL/6 mice,4T1-bearing-BALB/c mice	[94]
Tregs and MDSCs	sunitinib	polymeric micelle nano-delivery system (SUNb-PM) to increase cytotoxic T-cell infiltration and decrease the percentage of MDSCs and Tregs in the TME	C57BL/6 mice bearing B16F10 tumors	[97]
Tregs	imatinib	tLyp1 peptide-modified hybrid NPs, downregulate Tregs suppression and elevate intratumoral CD8^+^ T cells	C57BL/6 mice bearing B16/BL6 tumors	[98]
	-	CpG NPs/IR820-hydrogel, combined photothermal-immunotherapy by the dual fluorescence imaging method without additional fluorescent labeling	BALB/c mice, B16 cell line	[99]
Other immune molecules	-	NLG919@DEAP-^D^PPA-1 NPs, concurrent blockade of immune checkpoints and tryptophan metabolism	B16-F10 bearing mice model	[104]
	siRNA	LPH NP delivery TGF-β siRNA, increase tumor infiltrating CD8^+^ T cells and decrease Tregs	C57BL/6mice, B16F10 melanoma cell	[105]
	-	CSPM@CpG and synergistic PTT/PDT/immunotherapy	4T1-tumor-bearing mice model	[106]
3. CAF				
	-	Au@Ag NP, attenuate the tumor cell-promoting behavior of CAFs	NIH/3T3, MRC-5 fibroblast cells	[108]
	fraxinelloneand vaccine	nanoemulsion deliver fraxinellone and tumor-specific peptide vaccine, enhance anti-fibrosis ability and tumor-specific T-cell infiltration	Famale C57BL/6 mice, Murine BRAF-mutant melanoma cell line BPD6	[109]
4. ECM				
	pirfenidone	MRPL-PFD, downregulate ECM levels and enhance penetration of therapeutic drugs	Mia-paca-2 co-implanted tumor-bearing mice model	[112]
5. Exosomes				
	Pt prodrug	exosome capsule Pt(lau)-HSA-lecithin NPs develop chemotherapy for breast cancer	4T1 tumor bearing lung metastasis mice model	[119]
	-	AC electrokinetic direct immunoassay procedure permits specific identification and quantification of target biomarkers within as little as 30 min total time	Blood samples from patients	[120]
	-	magnetic beads conjugated with CD63 antibody for early diagnosis of cancer exosome	MDA-MB-231 cell line	[121]
6. Vasculature				
	cisplatin prodrug	cyclic pentapeptide and Pt (IV) loaded PLGA- PEG NPs targetingα(v)β(3) integrin were more efficacious and better tolerated	Female nude mice, DU145, MCF-7 cell line	[127]
	-	poly (acrylic acid) (PAA)-modified NaLnF4:40Gd/20Yb/2Er nanorods, for in vivo optical-guided tumor vessel/metastasis and noninvasive brain vascular imaging	LLC tumor bearing mice model	[128]
	-	anti-hVEGF molecularly imprinted polymer nanoparticles coupled with quantum dots for cancer imaging	WM-266 hVEGF(+) and A-375 hVEGF(−) model	[129]
	-	aggregation-induced emission (AIE) photosensitizer to fabricate integrin α(v)β(3) for image-guided and PDT	Nude mice, QBC939, L-O2, and HK-2 cells	[130]
7. Multiconstituents	-	Ca^2+^ in NPs can promote the maturation of DCs and release DAMPs from tumor cell to enhance infiltration of CTLs		[131]
	anti-CD47 and anti-SIRPα	exosomes NPs from M1 macrophages stopped SIRPα—CD47 axis in the acidic TME abolished the “don’t eat me” signal between tumor cell and macrophages and reprogram the M2 macrophages to M1 type	4T1 tumor-bearing BALB/c mice	[132]

## Data Availability

Not applicable.

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
