# Peer review of "Exploiting a New Approach to Destroy the Barrier of Tumor Microenvironment: Nano-Architecture Delivery Systems"

_molecules, 2021, doi:10.3390/molecules26092703_

Round 1

Reviewer 1 Report

The work by Sun and colleagues appears as a good contribution, rightly including some information about the new challenges about tumor therapies. Particular attention focuses on the TME and the role that this mixture of components has in opposition to the common chemotherapy approaches.

In my opinion, there are no important modification to do in order to make this work ready for publication.

These modification are about the organization of the paragraphs in the review. Since the 2.1 and the 2.2 section describe the major constituents and then the physiological conditions of the TME, respectively, the same order has not been kept for the point 3 that the authors divided into “3.1 Physiological condition mediated nanoparticles in TME” and “3.2 Major composition mediated nanoparticles in TME”. I think that the same organization in the section 2 and 3 has to be retain in order to make more flowing the text.

Author Response

Response to Reviewer 1 Comments

The work by Sun and colleagues appears as a good contribution, rightly including some information about the new challenges about tumor therapies. Particular attention focuses on the TME and the role that this mixture of components has in opposition to the common chemotherapy approaches.

In my opinion, there are no important modification to do in order to make this work ready for publication.

Point 1: These modifications are about the organization of the paragraphs in the review. Since the 2.1 and the 2.2 section describe the major constituents and then the physiological conditions of the TME, respectively, the same order has not been kept for the point 3 that the authors divided into “3.1 Physiological condition mediated nanoparticles in TME” and “3.2 Major composition mediated nanoparticles in TME”. I think that the same organization in the section 2 and 3 has to be retain in order to make more flowing the text.

Response 1: Thank you very much for your suggestion. We have adjusted the structure of section 3 in the revised manuscript. The organization in section 3 is consisted with section 2 now.

Reviewer 2 Report

The authors present a summary of the latest anti-cancer therapies using nanoparticles. The manuscript clearly introduces the reader in the first and second chapter to the most important issues related to TME. Chapter 3 summarizes the therapy with the use of nanotechnology. The work should be published because it summarizes in a new way the current research on nanotechnology therapies. Some elements are missing from the manuscript and some parts need to be expanded.

1) The authors correctly enter the abbreviations in the text, but the figure e.g. 2 -lack the expansion of abbreviation. This is the reason for their lower legibility. Please provide a list of abbreviations used throughout the whole manuscript.

2) NPs nanoparticles should be added to the list of key words

3) the use of “and so on” in the manuscript is not correct. Please delete this wording from the abstract and the text of the manuscript

4)  The authors introduce the reader to TME very well, but reader is not introduced to the possibilities of nanotechnology and nanoparticles. This should be added because the works cover both issues

5) In Figure 1, the phrase targeting multiconstituents does not contribute anything, please specify

6) In the description of immune cells, for all cells, except for dendritic cells, the percentages are given. Please complete this, possibly on the basis of also other works

7) Please explain more about the meaning of SIRPα in the context of M2 macrophages

8) Please explain more about the importance of MDSCs in non-response to treatment. One publication is cited 24. Are there any other works on this issue?

9) The topic of the inverse Warburg effect and CAFs supporting changes in cell metabolism has not been addressed in this paper. Please extend the work to include this. Are there treatments available with NPs to block the changes in metabolism of CAFs and cancer cells?

10) In the section 2.2.2 there is no information on the activation of mitochondrial biogenesis, which is enhanced by HIF1α and related to the amount of ATP

11) Figures 6 and 7 contain too many elements. They are too small, eg IF images, which are unreadable. Please change it or just make full-page format

I find this manuscript interesting and after making changes, it deserve to be published.

Author Response

Response to Reviewer 2 Comments

The authors present a summary of the latest anti-cancer therapies using nanoparticles. The manuscript clearly introduces the reader in the first and second chapter to the most important issues related to TME. Chapter 3 summarizes the therapy with the use of nanotechnology. The work should be published because it summarizes in a new way the current research on nanotechnology therapies. Some elements are missing from the manuscript and some parts need to be expanded.

Point 1: The authors correctly enter the abbreviations in the text, but the figure e.g. 2 -lack the expansion of abbreviation. This is the reason for their lower legibility. Please provide a list of abbreviations used throughout the whole manuscript.

Response 1 : Thank you for your advice. Your comments are very professional indeed. We have provided a list of abbreviations in section 5 in the revised manuscript.

Point 2: NPs nanoparticles should be added to the list of key words

Response 2 : Thank you for your suggestion. Nanoparticles is added to the key words list in the revised manuscript.

Point 3:  the use of “and so on” in the manuscript is not correct. Please delete this wording from the abstract and the text of the manuscript

Response 3 : Thank you for your comments. The phrase "and so on" in abstract and text has been deleted according to your kind suggestion in the revised manuscript.

Point 4: The authors introduce the reader to TME very well, but reader is not introduced to the possibilities of nanotechnology and nanoparticles. This should be added because the works cover both issues.

Response 4 : Thank you for your comments. Your advice is very professional. In the second paragraph of the introduction, we have added the advantages of nanotechnology and nanoparticles for drug delivery. Some organic nanomedicines which have been used in clinic and the characteristics of different inorganic nanomedicines are expounded too.

Point 5: In Figure 1, the phrase targeting multiconstituents does not contribute anything, please specify.

Response 5 : Thanks for your comments. We did not directly point out NPs targeting multiconstituents in the manuscript, so we made an explanation in section 3.1.6 in the revised manuscript.

Point 6: In the description of immune cells, for all cells, except for dendritic cells, the percentages are given. Please complete this, possibly on the basis of also other works.

Response 6 : Thanks for your comments. We read more references and found that DCs showed very small amounts even no DC cells were detected in some tumor tissues. At the same time, we found that the proportion of immune cells in different tumors was different. Therefore, in order to make the description more accurate and scientific, we only provided the classification of immune cells without percentages in the revised manuscript. We hope that you will be satisfied with this change.

Point 7: Please explain more about the meaning of SIRPα in the context of M2 macrophages

Response 7 : Thanks for your comments. We added the explanation of SIRPα and the role of CD47-SIRPα axis of phagocytosis in section 2.2.1. We hope that this explanation would bring some help to understand the content better.

Point 8: Please explain more about the importance of MDSCs in non-response to treatment. One publication is cited 24. Are there any other works on this issue?

Response 8 : Thanks for the comments. After reading more references, we added more descriptions about immunosuppressive functions of MDSCs in section 2.1.1 in the revised manuscript.

Point 9 : The topic of the inverse Warburg effect and CAFs supporting changes in cell metabolism has not been addressed in this paper. Please extend the work to include this. Are there treatments available with NPs to block the changes in metabolism of CAFs and cancer cells?

Response 9 : Thanks for your comments. We read more references about CAFs and elaborated their role in tumor metabolism especially glycolysis in section 2.1.3. It is a great pity that we have not found any NPs targeting CAFs that block the changes in metabolism for cancer treatment yet. We will follow up this relevant study in the future. Your suggestions will be greatly appreciated.

Point 10 : In the section 2.2.2 there is no information on the activation of mitochondrial biogenesis, which is enhanced by HIF1α and related to the amount of ATP

Response 10 : Thank you for your comments. The comments of reviewers are professional, the amount of ATP in tumor cells is produced by mitochondria. Aerobic glycolysis in tumors makes a lot of contributes to this result. Meanwhile, glycolysis is enhanced by HIF1 and other pathways. Thanks for your suggestion, we have modified the description in section 2.2.2.

Point 11 : Figures 6 and 7 contain too many elements. They are too small, eg IF images, which are unreadable. Please change it or just make full-page format.

Response 11 : Thank you for your comments. We changed Figure 6 and 7 to full-page format to enhance the legibility in the revised manuscript.